# Investigation of Fused Alumina Based-Mold Facecoats for DZ22B Directionally Solidified Blades

**DOI:** 10.3390/ma12040606

**Published:** 2019-02-18

**Authors:** Fei Li, Hongjun Ni, Lixiang Yang, Yi Jiang, Donghong Wang, Baode Sun

**Affiliations:** 1Shanghai Key Laboratory for High Temperature Materials and Precision Forming, Shanghai Jiao Tong University, Shanghai 200240, China; yanglixiang@sjtu.edu.cn (L.Y.); MarzPicture@sjtu.edu.cn (Y.J.); 2State Key Laboratory of Metal Matrix Composites, Shanghai Jiao Tong University, Shanghai 200240, China; bdsun@sjtu.edu.cn; 3School of Mechanical Engineering, Nantong University, Nantong 226019, China

**Keywords:** DZ22B nickel based superalloy, directionally solidified blade, mold facecoat, sand burning, investment casting

## Abstract

The interaction between alloy melt and mold facecoats is the main factor affecting the surface quality of investment casting nickel-based superalloys. An investigation was undertaken to develop suitable refractories as facecoat materials for the directionally solidified blades of DZ22B nickel-based superalloys in order to avoid a sand-burning defect. The wettability and interface reactions between alloy melt and various facecoats were studied by using a sessile drop experiment and the real investment casting method, respectively. The results show that by adding Cr_2_O_3_ powder with the amounts of 2 wt.%, 5 wt.% and 10 wt.% in the fused alumina-based facecoats, the wetting angles between the alloy melt and facecoats decreased from 105.40° to 100.37°, 99.96° and 98.11°, respectively, while the sand-burning defect on the casting blade surfaces still formed during the process of directional solidification. However, by adding h-BN powder in the fused alumina-based facecoats, the wetting angles between the alloy melt and facecoats dramatically increased, the sand-burning defect on the casting blade surfaces was effectively inhibited and a metallic luster on the directionally solidified blades could be obviously observed. In this study, the suitable composition of mold facecoats for the investment casting of blades is 2 wt.% h-BN added fused alumina.

## 1. Introduction

Nickel-based superalloys have been extensively adopted for producing turbine blades with excellent high-temperature tensile strength, stress rupture and creep properties in the fields of aero engines and gas turbines [1,2,3,4,5]. The directionally solidified investment casting process, a technique of solidification in accordance with the required crystalline orientation, is widely used for manufacturing superalloy blades with critical and complex details as well as precision dimensional tolerances [6,7,8,9,10,11,12]. The ceramic mold obtained by repeatedly dipping slurry and stuccoing plays an important role in the preparation of directionally solidified blades [13,14,15,16,17]. In the process of directional solidification, alloy melt is poured into the mold at high temperature. The interaction of thermo-mechanical penetration and thermophysical chemistry between the mold and the alloy melt occurs easily. During directional solidification, the alloy stays liquid for tens of minutes to several hours and the components of the mold material can easily react with the alloy melt [18,19]. Therefore, directionally solidified mold refractories should not only have a high softening point and low impurity content but also have good chemical stability to prevent a chemical reaction between the mold and alloy melt.

The sand-burning defect is one of the common defects of the components manufactured by investment casting, which is caused by the permeation of alloy melt into the mold facecoat or the interfacial reaction between the alloy melt and mold facecoat at high temperature [20,21]. The permeation between the alloy melt and mold facecoat is closely related to the wettability between the melt and mold, which is mainly determined by their compositions [22]. The equation for characterizing the wettability of liquids on solid surfaces is expressed as follows:(1)$$\cos\theta =(\sigma_{sl}-\sigma_{sg})/\sigma_{lg}$$
where θ is the wetting angle $\mathrm{and}\;\sigma_{sl}$, $\sigma_{sg}$ and $\sigma_{lg}$ are the surface tensions of liquid metal/vapor, ceramic/vapor and liquid metal/ceramic substrate, respectively. A smaller wetting angle corresponds to a better wettability between liquid and solid. It can be seen from Equation (1) that as the result of the interaction between the melt and mold facecoat, either mechanical or chemical sand-burning defects are directly related to the wettability between them. Infiltration of the alloy melt through the capillaries on the mold facecoat will easily occur if the alloy melt wets the mold facecoat well, which might lead to an obvious interfacial reaction. In recent years, research into the wettability between alloy melts and ceramic materials has been widely carried out. Chen et al. [23] have reported that with the increasing of C and Hf in an Ni_3_Al-based superalloy, the wetting angles of alloy melt on ceramic molds decrease and the interfacial reaction becomes more aggravated. Valenza et al. [24] have studied the wettability and interfacial reactions of three superalloys on different ceramic substrates and found that the compositions of alloy and ceramic have an influence on the wettability. Li et al. [22] have studied the wettability of TiAl alloy melt on ceramic molds in an electromagnetic field and found that the higher the ceramic chemical stability, the longer the nonreactive wetting time. Wang et al. [25] have studied the mechanisms of high-temperature wettability and interactions between NbSi-based alloys and Y_2_O_3_ ceramics and found that there was a characteristic transition in the wettability between the molten alloys and ceramics when the Y_2_O_3_ microstructure changed. From the above reports, it can be inferred that adjusting the wettability between the alloy melt and mold facecoat might be an effective method to inhibit the sand-burning defect of directionally solidified blades.

Alumina is one of the most widely used refractories in the directional solidification of superalloy blades [26,27]. It has stable high-temperature mechanical properties, a low thermal expansion coefficient and a relatively high chemical inertia to most alloy melts. However, in the process of preparing directionally solidified blades containing active elements, such as DZ417G, DZ22 and DD6, serious sand-burning defects have been found on the blade surfaces. Some researchers [20,21,28] have studied the interfacial reactions between alloy melts and fused alumina mold facecoats and preliminarily discussed the formation mechanism of the sand-burning defect. However, the methods to solve the sand-burning defect of directionally solidified superalloy blades have rarely been reported in the previous literature.

In this study, a simple and economical method to inhibit the sand-burning defect of nickel-based superalloy blades was developed by adding a special ceramic powder into the fused alumina mold facecoat. The interaction between the mold facecoat and the DZ22B nickel-based superalloy melt was investigated using energy dispersive spectrometry (EDS) and scanning electron microscopy (SEM). The wetting angles between the alloy melt and various mold facecoats were compared. The effects of the facecoat material properties and wettability between the alloy melt and mold facecoat on the sand-burning defect of the DZ22B blade were also discussed.

## 2. Materials and Methods

DZ22B nickel-based superalloy (C 0.14, Cr 9.00, Co 9.50, W 12.00, Nb 0.90, Ti 1.90, Al 4.90, Hf 1.00, B 0.015 and Ni as balance, wt.%) was used in this study. This alloy has the advantages of low density, high medium-temperature strength, good creep property, high plasticity, stable tissue and no valuable elements [29]. It is suitable for making turbine blades and guiding blades and other high-temperature parts when working under 980 °C.

Fused alumina powder (α-Al_2_O_3_ ≥ 99 wt.%, 325 mesh, applied by Hunan Lirong New Materials Co., Ltd., Changsha, China) was selected as a raw facecoat refractory and silica sol binder (1130C, applied by Nalco Company, Naperville, IL, USA) as binder. Cr_2_O_3_ powder (AR, 1~5 μm, applied by National Drug Group Chemical Reagent Beijing Co., Ltd., Beijing, China) and h-BN powder (99 wt.%, 10~30 μm, applied by National Drug Group Chemical Reagent Beijing Co., Ltd., Beijing, China) were used as additives in the fused alumina facecoat slurries. Latax (HX101, applied by Huzhou Long Tong Chemical Co., Ltd., Huzhou, China), wetting agent (Nalco 7667, applied by Nalco Company, Naperville, IL, USA), anti-foam agent (Nalco 2305, applied by Nalco Company, Naperville, IL, USA) and distilled water (self-preparation) were also added in these slurries in order to improve the slurry properties. The specific formulations of those slurries used for preparing the mold facecoats were designed and are given in Table 1.

The molds for the DZ22B superalloy directionally solidified blades were composed of a facecoat, transition coat and backup coats. Firstly, a wax combined with a gating system was dipped into the primary slurry and then stuccoed with 80-mesh fused alumina sand (α-Al_2_O_3_ ≥ 99 wt.%, applied by Sanmenxia Fused Alumina Co., Ltd., Sanmenxia, China). Secondly, after drying at 23 °C and 60% relative humidity for 24 h, the transition and backup coats using the backup slurry with the composition listed in Table 2 were performed with 46-mesh (for the transition coat) and 24-mesh (for the 3 backup coats) fused alumina sands as stucco materials. Lastly, the same backup slurry was applied without stuccoing. The transition coat and each backup coat were dried at a temperature of 23 °C, 60% relative humidity and 3 m/s air speed for 8 h. Steam autoclaving at 8 bar pressure for 15 min, followed by a controlled de-pressurization cycle at 1 bar/min, was performed for the removal of the wax inside the molds. After sintering at 850 °C in air for 2 h, the molds that could be used to prepared the directionally solidified DZ22B blades were obtained. Figure 1 shows the photos of the molds with different facecoats for the directional solidification of DZ22B alloy blades.

A Bridgman industrial vacuum furnace with heating zones and a withdrawal chamber was used to fabricate the DZ22B alloy blades by the directional solidification process. Firstly, after placing the mold on the water-cooled copper chill in the heating zone of the furnace chamber, the furnace was evacuated to 0.1 Pa and heated to 1500 °C. Secondly, the DZ22B alloy melt in a crucible was poured into the blade mold. Finally, the mold containing the alloy melt was withdrawn at the speed of 3.5 mm/min and the alloy was directionally solidified.

The micrographs of the refractories used in this study were observed by a scanning electron microscope (SEM, VEGA 3 Tesken, Czech Republic). The rheological properties of various primary slurries were measured with a rotational rheometer (Gemini 200HR, British Malvern Instruments Co., Ltd., Malvern, UK). The microstructures and compositions of the mold facecoats and interfaces between alloy and facecoats were analyzed using a scanning electron microscope (SEM, JMS-7800F Prime, Hitachi Company, Tokyo, Japan) and energy dispersive spectroscopy (EDS) equipped in SEM. The blades and ceramic molds were photographed by a digital camera.

The wettability of DZ22B alloy on the mold facecoat were investigated by a sessile drop test (Figure 2). The alloy samples with a size of 2 mm × 2 mm × 2 mm were placed on the mold facecoats, which had to be adjusted to a horizontal position on the cooling platform in the directional solidification furnace. The alloy samples on the ceramic substrate were heated to 1500 °C and melted completely (maintained for 20 min) in a vacuum condition (0.01 Pa). After the alloy/ceramic couples were cooled down to room temperature, the wetting angle (θ) was then calculated by using the geometric parameters *h* and *d* of the solidified alloy droplets, where *h* means the drop height and *d* is the base diameter of the alloy drop. The relationship expression between *h*, *d* and the wetting angle is θ=2arctan(2h/d), which is deduced from Equations (2)–(4). The cooled alloy dropped on the ceramic and the schematic drawing of the measurement of *h* and *d* are shown in Figure 3a,b respectively and the expression of *θ* is deduced from the following equations:(2)$$\tan\beta =\frac{h}{r}$$
(3)$$\beta =\arctan\frac{h}{r}=\arctan\frac{2h}{d}$$
(4)$$\theta =2\arctan\frac{2h}{d}$$

## 3. Results

The particle morphologies of the refractory powders used for preparing the primary slurries are shown in Figure 4. It can be seen from Figure 4a that the fused alumina powder is not uniform and consists of small (several micron) and large (tens of micron) particles with irregular shapes. Such a particle size distribution is beneficial to improve the facecoat compactness. As shown in Figure 4b, the Cr_2_O_3_ powder exhibits a uniform particle size distribution and most of the particles are less than 5 microns. Meanwhile, the shape of the Cr_2_O_3_ particles is close to spherical. From Figure 4c, it can be seen that the h-BN particles show an obvious flake-like morphology due to the hexagonal crystalline structure.

The directional solidification process for DZ22B nickel-based superalloy blades is over 45 min; thus, the mold has to sustain the alloy melt for a long time without deformation and cracking at high temperature. Meanwhile, the facecoat should be combined with the backup coats as tightly as possible in order to avoid peeling off during the casting procedure. Suitable mold sands can greatly improve the combination between the dry and fired facecoat and backup coats, as well as all mechanical properties of the mold. In this study, fused alumina sands with 80-mesh, 46-mesh and 24-mesh sizes were selected for the mold facecoat, transition coat and backup coats, respectively. Figure 5 shows the morphologies of the fused alumina sands observed by SEM. It can be seen from Figure 5 that both small and large sizes of fused alumina sands exhibit irregular morphology with a sharp angle structure but no fine sand particles are distributed among them. The fused alumina sands with such a shape can be easily anchored in the slurry coatings to improve the mechanical strength of the molds after drying and sintering.

The rheological properties of the primary slurries with various facecoat refractories have been characterized in order to compare the suitability of the primary slurries for the facecoat dipping process. Figure 6 presents the viscosity as a function of the shear rate at a constant temperature of 25 °C for fused alumina slurry and fused alumina slurries with various amounts added of Cr_2_O_3_ and h-BN, respectively. All the slurries for the rheological test have the same solid loading of 77 wt.%. As shown in Figure 6, all the slurries exhibited shear-thinning (pseudoplastic) behavior at a low shear rate and shear-thickening (dilatant) behavior at a high shear rate. The shear-thickening behavior is due to the large inter-particle electrostatic repulsive forces existing in the slurries. In comparison with the fused alumina slurry, the fused alumina slurries with added Cr_2_O_3_ have lower viscosity, while that with added h-BN exhibits a higher viscosity at the same shear rate. It should be noted here that, as the solid loading was fixed, the viscosity of the Cr_2_O_3_ added slurry decreased with the increasing of the Cr_2_O_3_ amount and the h-BN added slurry increased with the increasing of the h-BN amount. From the above results, it can be deduced that adding Cr_2_O_3_ powder in fused alumina slurry is beneficial to improve the slurry fluidity due to the spherical shape of the Cr_2_O_3_ particles, while adding h-BN powder is harmful to the slurry fluidity due to the flake-like structure of h-BN. It is also found from the real experiment that, when the added amount of h-BN in the slurry was 5 wt.% ratio to fused alumina, the slurry became too thick to achieve uniform coating on the surface of the whole blade wax pattern.

The surface station of the facecoat has a great effect on the interaction between the alloy melt and ceramic mold; thus, it is necessary to improve the facecoat properties in order to guarantee the surface quality of the castings. Figure 7 shows the surface microstructure of the fused alumina facecoat. As shown in Figure 7, some pores with micron sizes were distributed on the facecoat of the fused alumina mold that has been fired at 850 °C for 2 h. The existence of the pores in the facecoat can improve the permeability of the ceramic mold but excessive pores lead to the penetration of the melt into the mold facecoat, which might increase the risk of thermal mechanical sand burning on the directionally solidified blade. Figure 8 shows the surface microstructures of the fused alumina facecoats with added Cr_2_O_3_ powder. It can be seen from Figure 8 that the Cr_2_O_3_ added facecoats become denser in contrast of the fused alumina facecoat (Figure 7). This indicates that the fine spherical Cr_2_O_3_ particles are likely to fill the pores between the fused alumina particles. Interestingly, when the additive content of Cr_2_O_3_ is 10 wt.% ratio to fused alumina, a large number of tiny Cr_2_O_3_ particles could be observed on the surface of mold facecoat, as shown in Figure 8c. Figure 9a,b show the surface microstructures of fused alumina facecoats with added 2 wt.% and 5 wt.% h-BN powder (ratio to fused alumina). It can be seen from Figure 9 that the h-BN added facecoats exhibited similar microstructures to that of the fused alumina facecoat but the quantity of pores between the ceramic particles seemed to be smaller.

The melt penetration usually takes place when the melt wets the facecoat well. Here, the sessile drop tests of the DZ22B superalloy melt on various facecoats at 1500 °C were measured as shown in Figure 10 and the results are listed in Table 3. As shown in Table 3, the wetting angle of the fused alumina/melt couple was 105.40°, while by adding 2 wt.%, 5 wt.% and 10 wt.% Cr_2_O_3_ in the fused alumina facecoats, the wetting angle decreased to 100.37°, 99.96° and 98.11°, respectively, which means that Cr_2_O_3_ additive could increase the wettability of alloy melt on fused alumina facecoats. However, by adding h-BN powder in fused alumina facecoats with the 2 wt.% and 5 wt.% ratio to fused alumina, the wetting angles dramatically increased to 114.19° and 116.05°, which indicates that h-BN could decrease the wettability of alloy melt on fused alumina facecoats.

In order to compare the effect of Cr_2_O_3_ and h-BN additives on the modification of fused alumina facecoats with the same amount of addition, the directionally solidified blades were prepared by using the molds with fused alumina facecoats, 2 wt.% and 5 wt.% Cr_2_O_3_ added fused alumina facecoats as well as 2 wt.% and 5 wt.% h-BN added fused alumina facecoats, respectively.

Figure 11a,b show the macroscopic photo and cross-sectional SEM image of the as-casted DZ22B alloy blades by using ceramic molds with fused alumina facecoats. It can be seen that all the blade bodies are covered by a pinkish sand-burning layer with an uneven surface (Figure 11a) and the thickness of the layer is about 20 μm. The result of EDS analysis at positions 1, 2 and 3 indicates that the sand-burning layer mainly contains Al, Si, O and Cr elements, as shown in Table 4. According to the compositions of the mold facecoat and alloy, it can be deduced that Al, Si, O came from the facecoat, while Cr should come from the alloy. Some researchers [30] have reported that Cr in the alloy melt can react with impurities such as Fe_2_O_3_, Na_2_O in the mold to form highly active Cr_2_O_3_, which further reacts with alumina in the mold to form a pink solid solution at high temperature. According to the previous reports and the results obtained in this study, it can be inferred that the formation of a sand-burning defect on the DZ22B blade surface is partly related to the high-temperature interfacial reaction between the alloy melt and the fused alumina facecoat. In future work, XRD and XPS analysis will be carried out to accurately characterize the phase composition and chemical composition of the sand burning layer.

Figure 12 shows the macroscopic photos and cross-sectional SEM images of the as-casted DZ22B alloy blades by using ceramic molds with 2 wt.% and 5 wt.% Cr_2_O_3_ added fused alumina facecoats. It can be seen from Figure 12a,c that the sand-burning defect still appeared on the directionally solidified blades, although the fused alumina facecoats with added 2 wt.% and 5 wt.% Cr_2_O_3_ powder were much denser than the fused alumina facecoat (as shown in Figure 7 and Figure 8, respectively). Comparing with Figure 11b and Figure 12b, it can be seen that adding 2 wt.% Cr_2_O_3_ to the fused alumina facecoat could reduce the thickness of the blade sand-burning layer to less than 10 microns. However, increasing the content of Cr_2_O_3_ in the facecoat led to a more serious sand-burning defect. As shown in Figure 12d, the sand-burning layer of the blade prepared by using the 5 wt.% Cr_2_O_3_ added mold facecoat is thicker than that prepared by using the 2 wt.% Cr_2_O_3_ added mold facecoat. Moreover, many non-metallic inclusions (Figure 12d) formed near the interfacial area, which was definitely caused by the penetration of alloy melt into the mold facecoat during the blade casting procedure and then enwrapped the facecoat materials to form an inclusion defect after the alloy solidification. Table 5 shows the EDS analysis results of positions 1–3 in Figure 12b and 4–6 in Figure 12d. As expected, Al, O, Si and Cr elements have also been detected in the sand-burning layer of the blade prepared by using the mold with a Cr_2_O_3_ added facecoat. Moreover, the content of Cr in the sand-burning layer increases with the increase of Cr_2_O_3_ content in the mold facecoats. Meanwhile, Ni and Hf elements which came from the alloy were also detected. It is noted here that the sand-burning layer exhibits a grayish white color (Figure 12a,c) but not a pinkish color, which indicates that no solid solution was formed during the directionally solidified process by using the Cr_2_O_3_ added facecoat. This might due to the presence of the additional Cr_2_O_3_ in the mold facecoat inhibiting the diffusion of Cr from the melt to the alloy/facecoat interface during the directional solidification; thus, no solid solution with pinkish color formed.

Figure 13 shows the macroscopic photos and cross-sectional SEM images of the as-casted DZ22B alloy blades by using a ceramic mold with 2 wt.% and 5 wt.% h-BN added fused alumina facecoats. Macroscopically, no obvious sand burning can be observed either on the tenon parts or body parts of the DZ22B superalloy blades and metallic luster can be seen from the blades, as shown in Figure 13a. Contrastingly, slight sand-burning layers still appeared on part of the blades prepared by using the mold with a 5 wt.% h-BN added facecoat (Figure 13c). As mentioned above, it was difficult to uniformly dip the primary slurry composed of fused alumina + 5 wt.% h-BN on the blade wax due to its high viscosity and poor fluidity. Some positions on the blade, such as the blade basin and blade body near the tenon, were prone to slurry accumulation in the process of mold facecoat preparation. Thus, the sand-burning defect of the blades formed at such positions with low facecoat strength. It can be seen from Figure 13b,d that the interface thickness of the blade prepared by using the h-BN added facecoat is much thinner than that prepared by using fused alumina and Cr_2_O_3_ added facecoats. This indicates that h-BN addition in the facecoat dramatically inhibited the formation of the sand-burning defect on the DZ22B blade. It can be also found that some independent white particles formed, as shown at positions 2 and 4 in Figure 13b,d, which was characterized by EDS (Table 6) and attributed to HfO_2_ as reported by other researchers [21]. From the above study, it can be inferred that the mold with 2 wt.% h-BN added fused alumina facecoat is more suitable than others for the investment casting of DZ22B nickel-based superalloy blades.

## 4. Discussion

As reported as in the previous literature [21,23,25,30,31], some elements such as Cr, Hf and Al in the nickel-based alloy could promote the interface reaction between the alloy melt and mold facecoat. These elements also exist in large amounts in DZ22B nickel-based superalloys. The sand-burning defect is a long-term problem in the manufacture of directionally solidified DZ22B blades. In this study, it was found that many pores were distributed on the fused alumina facecoat, as shown in Figure 7. Whether the alloy melt can penetrate into the pores of the mold facecoat and cause thermo-mechanical penetration sand burning mainly depends on the ability of the melt to overcome the critical capillary force. The capillary force *P* is expressed as in Equation (5):(5)$$P=\frac{2\sigma \cos\theta }{r}$$
where *σ*—surface tension of alloy melt;

*θ*—wetting angle between alloy melt and mold facecoat;

*r*—radius of pore in mold facecoat.

Equation (5) indicates that the increase of the capillary force leads to the improvement of wettability between the alloy melt and mold facecoat. Thus, the melt becomes much easier to infiltrate into the pores of the mold facecoat and causes serious sand-burning defects on the DZ22B blades prepared by using a mold with a porous fused alumina facecoat, as shown in Figure 11. Adding Cr_2_O_3_ powder in the fused alumina facecoats could dramatically reduce the pores in those facecoats, as shown in Figure 8. However, with the contents of Cr_2_O_3_ increasing from 0 wt.% to 10 wt.%, the wetting angles between the alloy melt and mold facecoats decreased from 105.42° to 98.11°. This means that the wettability of the alloy melt on the surface of mold facecoats became much stronger with increased Cr_2_O_3_ addition. An obvious sand-burning defect could be still observed on the DZ22B blade surfaces and even some non-metallic inclusions appeared in the subsurface area of the blades when Cr_2_O_3_ was added to the mold facecoat at 5 wt.% to the fused alumina. The above results show that reducing the wettability between the melt and mold facecoat might be an effective method to inhibit the mechanical sand-burning defect on the directionally solidified DZ22B alloy blades.

The wettability of the DZ22B alloy melt on the fused alumina facecoat could be dramatically reduced by adding h-BN powder in the facecoat, as shown in Table 4. It is also well-known that h-BN, with a hexagonal crystal structure, is a high-temperature solid lubricant and is not wetted by most liquid metals. Therefore, h-BN is selected as a key additive and added to the fused alumina facecoat in order to reduce the wettability between the alloy melt and mold facecoat in this study. The decrease of wettability is beneficial to reduce the penetration of metal melt into the mold facecoat pores, thereby reducing the mechanical sand-burning defects on DZ22B blades.

From the above study, it can be concluded that adding h-BN powder can reduce the pore size of the facecoat and make the ceramic mold resist the wetting of metal melt. The penetration of alloy melt into the h-BN added facecoat becomes more difficult than the pure fused alumina facecoat and thus the sand-burning defect is dramatically inhibited. In this study, the suitable addition amount of h-BN in the fused alumina facecoat is only 2 wt.% to fused alumina. In addition, the price of this kind of ceramic mold with excellent sand-burning resistance is almost the same as that of the traditional mold.

## 5. Conclusions

This study proposed a new fused alumina based mold facecoat for preventing the DZ22B blades from experiencing a sand-burning defect in the directional solidification process. The following conclusions can be drawn:Serious sand-burning defects exist on the directionally solidified blades of DZ22B nickel-based superalloys. The formation mechanism of the sand-burning defect is attributed to the thermal mechanical permeation of alloy melt into the mold facecoat and the interfacial reaction between the alloy melt and mold facecoat.Adding Cr_2_O_3_ powder in the fused alumina facecoat can make the facecoat denser but greatly decreases the wetting angles between the alloy melt and mold facecoat. The thermal mechanical permeation of the alloy melt into mold facecoat is the main mechanism behind the sand-burning defect on DZ22B blades due to the strong wettability of alloy melt on mold facecoats with added Cr_2_O_3_.Adding h-BN in the fused alumina facecoat not only reduces the pore size in the facecoat but also decreases the wettability of the alloy melt on the mold facecoat. The penetration of alloy melt into the h-BN added facecoat becomes more difficult than the pure fused alumina facecoat and thus the sand-burning defect is dramatically inhibited. In this study, the suitable addition amount of h-BN in the fused alumina facecoat is only 2 wt.% to fused alumina.

## Figures and Tables

**Figure 1 materials-12-00606-f001:**
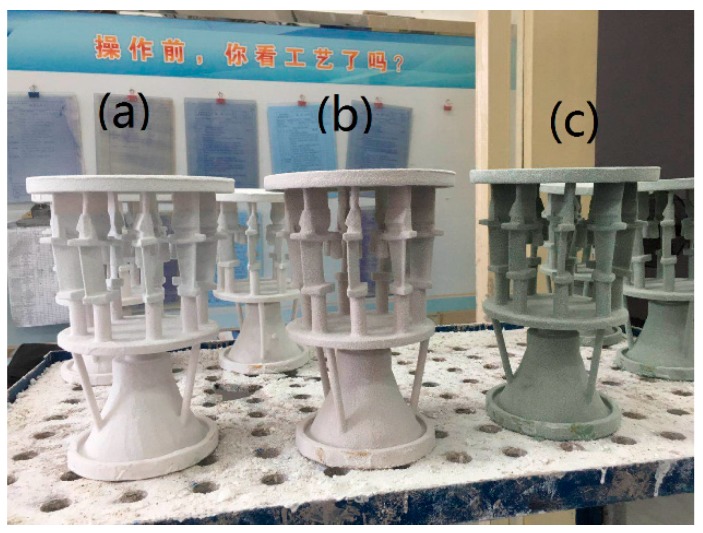
Photos of (**a**) fused alumina facecoat; (**b**) 2 wt.% h-BN added facecoat and (c) 2 wt.% Cr_2_O_3_ added facecoat.

**Figure 2 materials-12-00606-f002:**
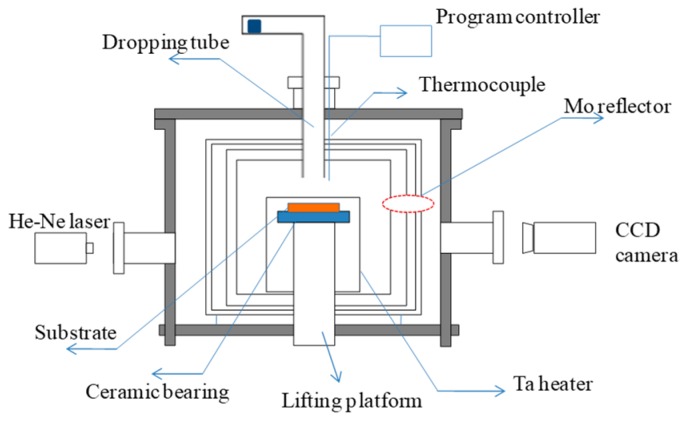
Schematic illustration of equipment used for the sessile drop experiment.

**Figure 3 materials-12-00606-f003:**
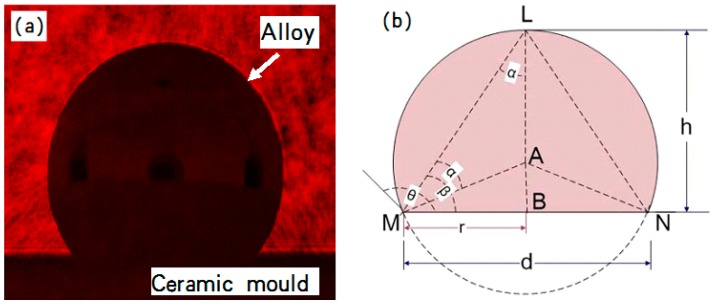
Images of (**a**) the alloy drop on the ceramic and (**b**) the schematic illustration of the measurement of *h* and *d*.

**Figure 4 materials-12-00606-f004:**
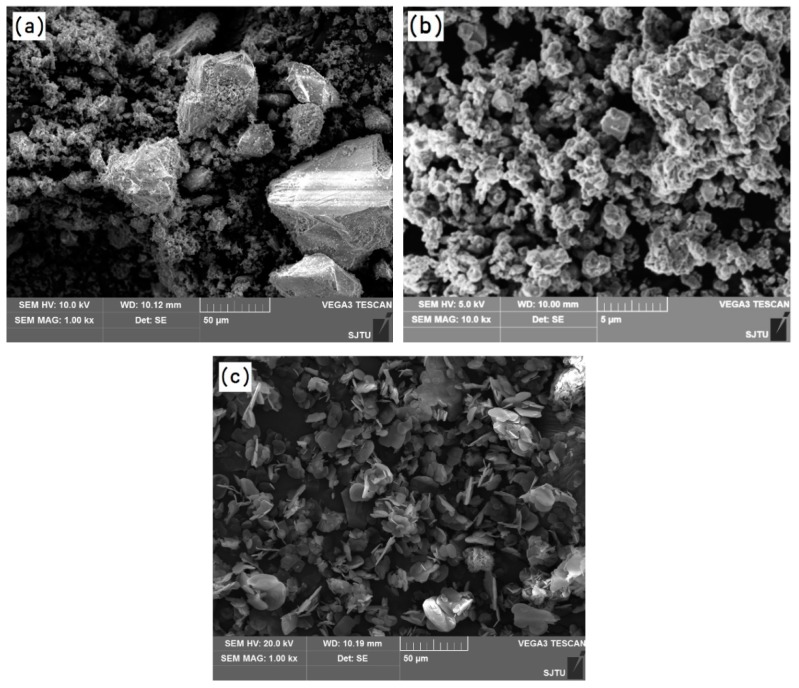
Particle morphology of the refractory powders: (**a**) fused alumina, (**b**) chromium oxide and (**c**) hexagonal boron nitride.

**Figure 5 materials-12-00606-f005:**
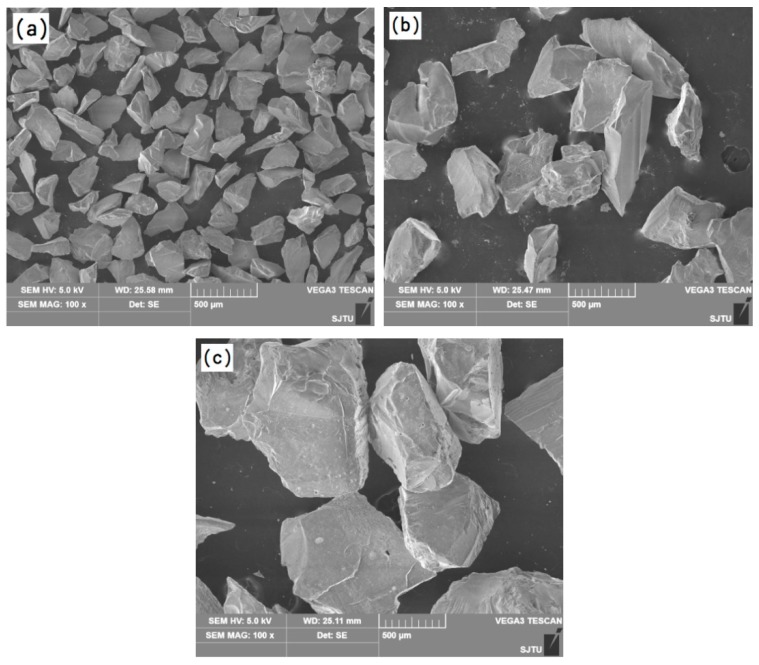
SEM micrographs of fused alumina sands: (**a**) 80-mesh; (**b**) 46-mesh and (**c**) 24-mesh.

**Figure 6 materials-12-00606-f006:**
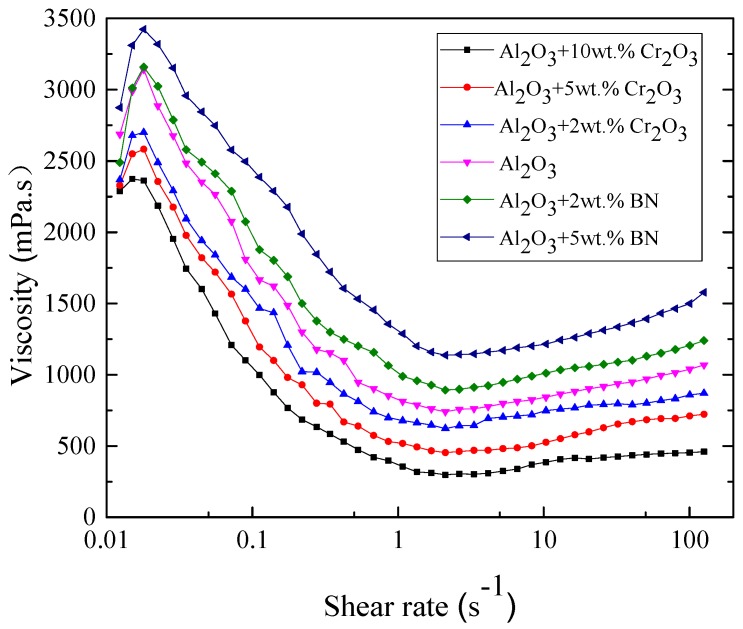
Viscosity as a function of shear rate for various primary slurries (solid loading: 77 wt.%).

**Figure 7 materials-12-00606-f007:**
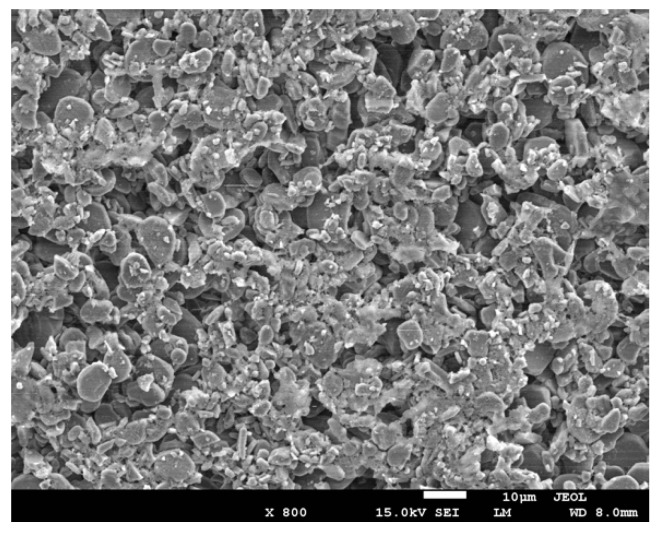
Microstructure of the fused alumina facecoat after fired at 850 °C in air for 2 h.

**Figure 8 materials-12-00606-f008:**
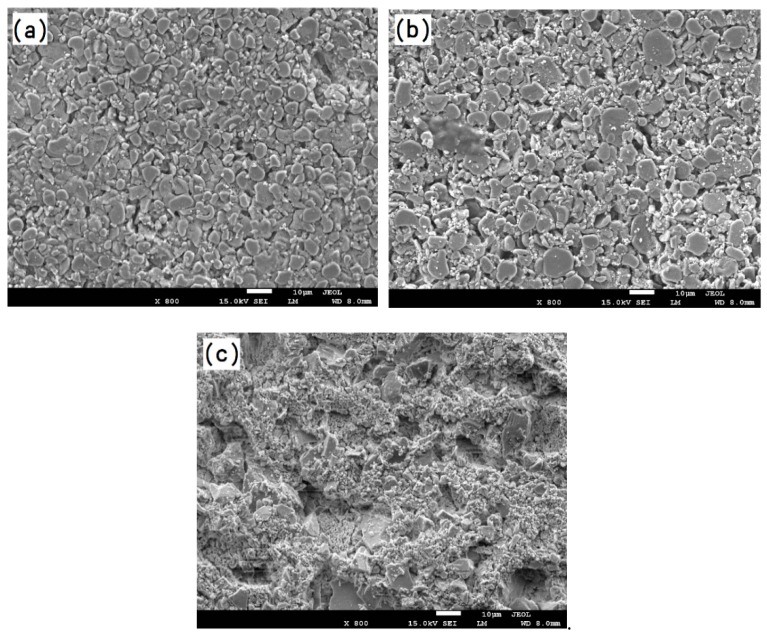
Microstructure of the fused alumina facecoats with added (**a**) 2 wt.% Cr_2_O_3_; (**b**) 5 wt.% Cr_2_O_3_ and (**c**) 10 wt.% Cr_2_O_3_ after fired at 850 °C in air for 2 h.

**Figure 9 materials-12-00606-f009:**
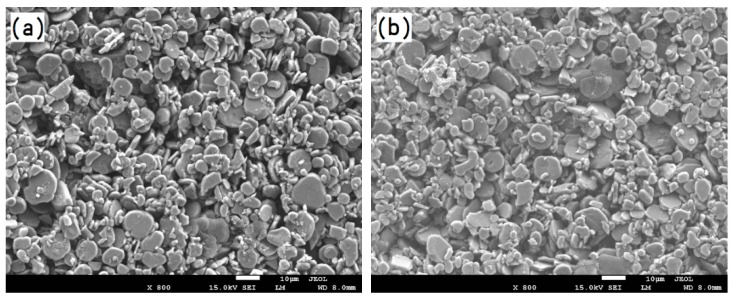
Microstructure of the fused alumina facecoats with added (**a**) 2 wt.% h-BN and (**b**) 5 wt.% h-BN after fired at 850 °C in air for 2 h.

**Figure 10 materials-12-00606-f010:**
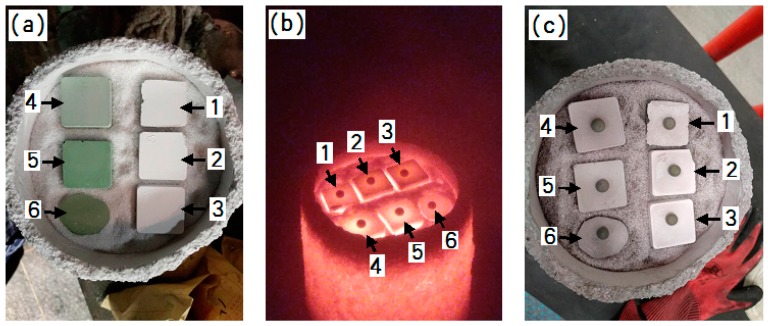
Mold facecoat samples for sessile drop test. (**a**) Before test; (**b**) during testing and (**c**) after test (1-fused alumina; 2-fused alumina + 2 wt.% h-BN; 3-fused alumina + 5 wt.% h-BN; 4-fused alumina + 2 wt.% Cr_2_O_3_; 5-fused alumina + 5 wt.% Cr_2_O_3_; 6-fused alumina + 10 wt.% Cr_2_O_3_).

**Figure 11 materials-12-00606-f011:**
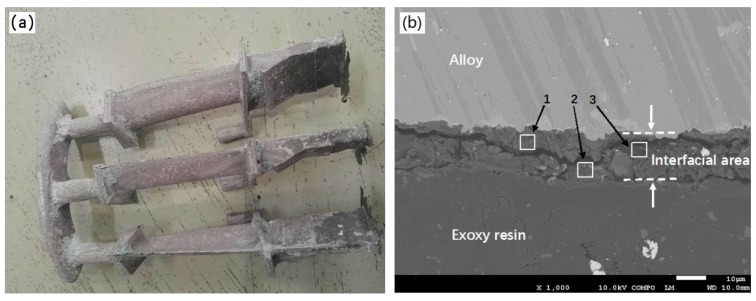
Macroscopical photos and cross-sectional SEM images of as-casted DZ22B alloy blades. (**a**) Blade photo (facecoat: fused alumina); (**b**) interfacial area image (facecoat: fused alumina).

**Figure 12 materials-12-00606-f012:**
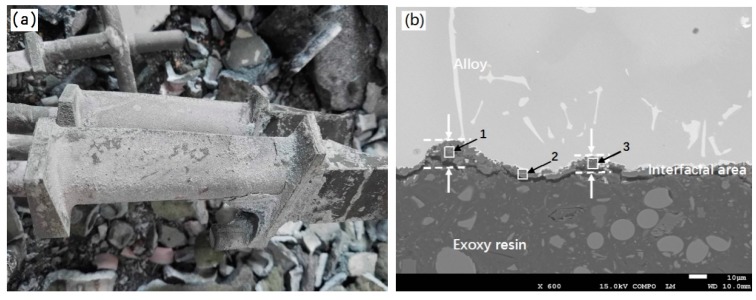
Macroscopical photos and cross-sectional SEM images of as-casted DZ22B alloy blades. (**a**) Blade photo (facecoat: fused alumina+ 2 wt.% Cr_2_O_3_); (**b**) interfacial area image (facecoat: fused alumina+ 2 wt.% Cr_2_O_3_); (**c**) blade photo (facecoat: fused alumina+ 5 wt.% Cr_2_O_3_) and (**d**) interfacial area image (facecoat: fused alumina+ 5 wt.% Cr_2_O_3_).

**Figure 13 materials-12-00606-f013:**
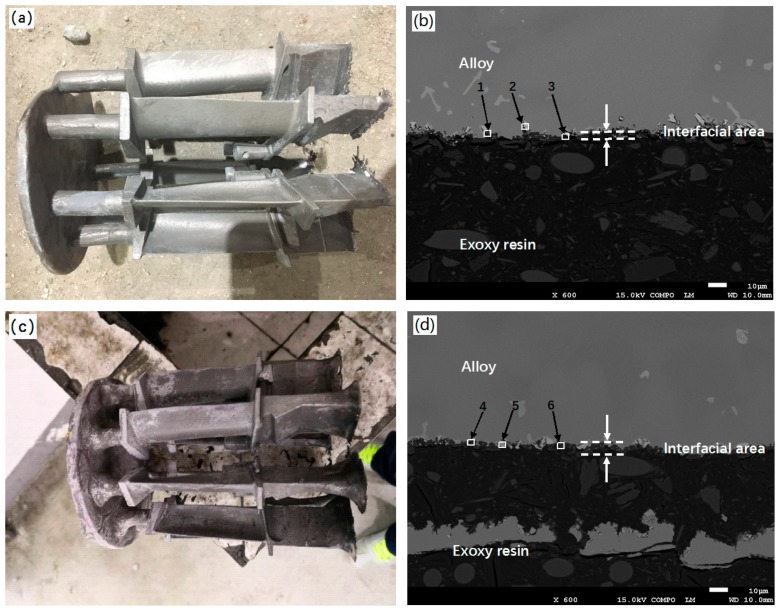
Macroscopical photos and cross-sectional SEM images of as-casted DZ22B alloy blades. (**a**) Blade photo (facecoat: fused alumina + 2 wt.% h-BN); (**b**) interfacial area image (facecoat: fused alumina + 2 wt.% h-BN); (**c**) blade photo (facecoat: fused alumina + 5 wt.% h-BN) and (**d**) interfacial area image (facecoat: fused alumina + 5 wt.% h-BN).

**Table 1 materials-12-00606-t001:** Primary slurry composition for ceramic molds.

Primary Slurries	Materials	Composition
Mold 1: Al_2_O_3_ slurry	Liquids: Silica sol, latex, wetting agent, anti-foam agent, distilled water Refractory: Al_2_O_3_ powder	23 wt.% of total liquids 77 wt.% of refractory loading
Mold 2: Al_2_O_3_+Cr_2_O_3_ slurry	Liquids: Silica sol, latex, wetting agent, anti-foam agent and distilled water Refractories: Al_2_O_3_ + Cr_2_O_3_ powders	23 wt.% of total liquids 77 wt.% of refractory loading (M_Cr2O3/_M_Al2O3_: 2 wt.%, 5 wt.% and 10 wt.%)
Mold 3: Al_2_O_3_+h-BN slurry	Liquids: Silica sol, latex, wetting agent, anti-foam agent and distilled water Refractories: Al_2_O_3_ + h-BN powders	23 wt.% of total liquids 77 wt.% of refractory loading (M_h-BN/_M_Al2O3_: 2 wt.% and 5 wt.%)

**Table 2 materials-12-00606-t002:** Backup slurry composition for ceramic molds.

Slurry	Materials	Composition
Backup slurry	Liquids: Silica sol, anti-foam agent, distilled water Refractory: Al_2_O_3_	33 wt.% of total liquids 67 wt.% of refractory loading

**Table 3 materials-12-00606-t003:** Wetting angles of alloy melt on the mold facecoats.

Samples	Composition of Mold Facecoats	Wetting Angle (°)
1	Fused alumina	105.42
2	Fused alumina + 2 wt.% h-BN	114.19
3	Fused alumina + 5 wt.% h-BN	116.05
4	Fused alumina + 2 wt.%Cr_2_O_3_	100.37
5	Fused alumina + 5 wt.%Cr_2_O_3_	99.96
6	Fused alumina + 10 wt.%Cr_2_O_3_	98.11

**Table 4 materials-12-00606-t004:** EDS analysis of the points in Figure 10b (at.%).

Position	Al	Si	O	Cr
1	36.43	2.10	58.05	5.42
2	33.32	1.40	59.98	8.30
3	33.71	1.24	66.05	-

**Table 5 materials-12-00606-t005:** EDS analysis of the positions in Figure 11b,d (at.%).

Position	Al	Si	O	Cr	Ni	Hf
1	38.41	2.10	46.75	10.97	1.26	0.51
2	37.60	1.40	47.21	9.66	2.18	1.95
3	36.73	1.24	49.88	10.21	2,94	-
4	36.11	1.13	46.12	12.41	-	4.23
5	33.85	0.98	47.19	14.34	1.49	2.15
6	37.42	0.73	41.48	15.98	1.11	3.28

**Table 6 materials-12-00606-t006:** EDS analysis of the positions in Figure 12b,d (at.%).

Position	Al	Si	O	N	Ti	Ni	Hf
1	23.45	1.58	28.98	1.15	0.11	42.15	2.58
2	4.36	0.99	39.54	0.38	-	0.22	54.51
3	40.91	1.65	43.44	1.29	-	11.48	1.23
4	44.19	2.12	34.12	2.19	0.23	9.98	7.35
5	15.15	1.46	39.65	2.05	-	0.47	41.22
6	47.59	1.15	31.44	2.37	0.12	11.12	6.21

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
