# Peer review of "Investigation of Fused Alumina Based-Mold Facecoats for DZ22B Directionally Solidified Blades"

_materials, 2019, doi:10.3390/ma12040606_

Round 1
Reviewer 1 Report
Dear Authors,
The work is very interesting, especially in the context of the application. The introduction contains sufficient content, supported by literature. Chapters of Materials and methods and Results were been developed sufficiently. However, before the publication, it's necessary to explain a few issues contained in the attachment.
Kind regard,
Reviewer

Author Response
Detailed response to Reviewer 1
Journal: Materials
Manuscript ID: materials-429784
Title: Investigation of fused alumina based mould facecoats for DZ22B directionally solidified blades
Authors: Fei Li *, Hongjun Ni* , Lixiang Yang , Yi Jiang , Donghong Wang *, Baode Sun
Dear reviewer,
Thank you for the valuable comments and suggestions on our manuscript. We have modified the manuscript accordingly, and highlight the precise revised changes in the paper. The detailed corrections are listed below point by point:
(1) Lines 100, 103, 105, 159 – for example: (…) with stucco of 80 mesh (…). Or maybe: fused alumina about graininess / gradation 80.
Answer: The granularity unit of fuse alumina sand we purchased is mesh (80-mesh for the facecoat, 46-mesh for the transition coat and 24-mesh for the backup coats, shown in Chapter 2 (Materials and Methods), Page 3 in the revised manuscript).
(2) Is the word mesh correctly used? 80 - How much is that mm? Is it sand or powder?
Answer: The word “mesh” is correctly used. The average particle sizes of the fused alumina sand corresponding to 80-mesh, 46-mesh and 24-mesh are about 0.180 mm, 0.355 mm and 0.710 mm, respectively, which can be seen in Figure 5 in the revised manuscript.
(3) How many samples were tested for each technique applied research?
Answer: Five samples were used to test the wetting angles for every facecoat and the average values were obtained in this study. Eight blades were obtained for every mould with different facecoat. One slurry was used to determine the rheological properties for every facecoat.
(4) Lines 240-242 – In the Fig. 12a not only blades have a metallic luster. Please explain it.
Answer: We can see from Figure 12a that all the blades prepared by using 2wt.% h-BN added fused alumina facecoat show a metallic luster and no obvious sand burning layer exist on the blade surface.
(5) Line 253 – This kind of information should be rather at the beginning of the publication.
Answer: We have rewritten this part, please see Chapter 4 for details in the revised manuscript.
(6) Lines 294-297 - This is to be a summary whether conclusions?
Answer: This part (line 294-297 in the first manuscript) have be rewritten as following:
This study proposed a new fused alumina based mould facecoat for preventing the DZ22B blades from experiencing a sand-burning defect in the directional solidification process. The following conclusions can be drawn:
(7) Lines 298-301 - This doesn't seem to be an application resulting from the conducted research.
Answer: Yes, this is not an application result, but can be used to explain the mechanism of sand burning defect of the blade prepared by using pure fused alumina facecoat (line 298-301 in the first manuscript). On this basis, we studied the effect of adding Cr2O3 and h-BN on inhibiting the sand burning defect of the DZ22B blades. Therefore, we take it as the first part of the conclusion as shown in the revised manuscript.
The manuscript has been resubmitted to the journal. We look forward to your positive response.
Sincerely,
Dr. Fei Li
1 Shanghai Key Laboratory for High Temperature Materials and Precision Forming, Shanghai Jiao Tong University, Shanghai 200240,China
2 State Key Laboratory of Metal Matrix Composites, Shanghai Jiao Tong University, Shanghai 200240, China
Email address: [email protected]
Tel/Fax: +86-21-34202951

Reviewer 2 Report
The manuscript is well-prepared and present small, but nicely presented, batch of results. My comments concern mostly the typing errors. My general regard is that it would be nice to use FT-IR or Raman spectroscopy to characterize the studied materials.
line 50: "It" instead of "it"
lines 50-54: this should be supported with the suitable citation or the relevant literature
lines 84-90: this should be supported with the suitable citation or the relevant literature
there should be a space between the "Figure" and its number, in multiple places in the article
there should be no space between the value of the angle and its symbol i.e. in the lines 206, 208
Figure 223: "shows"
lines 384-385: this extra space is not needed
Author Response
Detailed response to Reviewer 2
Journal: Materials
Manuscript ID: materials-429784
Title: Investigation of fused alumina based mould facecoats for DZ22B directionally solidified blades
Authors: Fei Li *, Hongjun Ni* , Lixiang Yang , Yi Jiang , Donghong Wang *, Baode Sun
Dear reviewer,
Thank you for the valuable comments and suggestions on our manuscript. We have modified the manuscript accordingly, and highlight the precise revised changes in the paper. The detailed corrections are listed below point by point:
line 50: "It" instead of "it"
Answer: “it” has been changed to “It” in the revised manuscript.
lines 50-54: this should be supported with the suitable citation or the relevant literature.
Answer: we have rewritten this part (Chapter 1 Introduction) and added some literatures (references 20-25) in the revised manuscript.
lines 84-90: this should be supported with the suitable citation or the relevant literature
Answer: we have rewritten this part (Chapter 1 Introduction) and added some literatures (references 20-25) in the revised manuscript.
there should be a space between the "Figure" and its number, in multiple places in the article
Answer: The space between the “Figure” and tis number has been added.
there should be no space between the value of the angle and its symbol i.e. in the lines 206, 208
Answer: The space between the value of the angle and its symbol i.e. in the lines 206, 208 has been deleted. Please see the revised manuscript.
Line 223: "shows"
Answer: “show” has been changed to “shows”. Please see the revised manuscript.
lines 384-385: this extra space is not needed
Answer: The extra space has been deleted (lines 384-385). Please see the revised manuscript.
Comments and suggestions for Authors: My general regard is that it would be nice to use FT-IR or Raman spectroscopy to characterize the studied materials.
Answer: This paper mainly focused on how to inhibit the sand burning defect on the directly solidified blades by changing the wettability between DZ22B superalloy melt and mould facecoat. The above research could basically explain this mechanism through comparing the wetting angles of the alloy melt on various facecoats and the thickness of the interface layers. In the future, we will fully consider using FT-IR or Raman Spectroscopy proposed by reviewers to characterize the materials.
The manuscript has been resubmitted to the journal. We look forward to your positive response.
Sincerely,
Dr. Fei Li
1 Shanghai Key Laboratory for High Temperature Materials and Precision Forming, Shanghai Jiao Tong University, Shanghai 200240,China
2 State Key Laboratory of Metal Matrix Composites, Shanghai Jiao Tong University, Shanghai 200240, China
Email address: [email protected]
Tel/Fax: +86-21-34202951

Reviewer 3 Report
The manuscript reports "Investigation of Fused Alumina Based Mold Facecoats for 2 DZ22B Directionally Solidified blades" however it does not present all the EDS spectra of each micrographs that are important because they varied the composition of Cr2O3 powder with the amounts of 2wt.%, 5wt.% and 19 10wt.%,
It is recommended to carry out the following evaluations:
- XRD pattern: it is suggested discussing crystallographic structure and variations.
-Elemental analysis through ICP or spark
- Authors have to point out the advantages of the proposed methodology, time, quantity of sample and costs compared to other methodologies by adding references.The citations are somewhat lacking in the discussion.
- In the Table 3 add a graph that facilitates the understanding of results
Minorpoints:
1. Line 34 Reference 1 does not talk about nickel alloys review
2. Line 41 Space in 30min ~ 90min, correct 30min ~ 90min
3. Line 46-59 Add references there are none
4. Line 60-64 Add references there are none
5. Table 4 homogenize decimals
6. Line 102 space 24h. correct 24 h. reviewing the entire document is consistent this error
Author Response
Detailed response to Reviewer 3
Journal: Materials
Manuscript ID: materials-429784
Title: Investigation of fused alumina based mould facecoats for DZ22B directionally solidified blades
Authors: Fei Li *, Hongjun Ni , Lixiang Yang , Yi Jiang , Donghong Wang *, Baode Sun
Dear reviewer,
Thank you for the valuable comments and suggestions on our manuscript. We have modified the manuscript accordingly, and highlight the precise revised changes in the paper. The detailed corrections are listed below point by point:
Comments and Suggestions for Authors
The manuscript reports " Investigation of fused alumina based mould facecoats for DZ22B directionally solidified blades", however it does not present all the EDS spectra of each micrographs that are important because they varied the composition of Cr2O3 powder with the amounts of 2wt.%, 5wt.% and 10wt.%, It is recommended to carry out the following evaluations:
XRD pattern: it is suggested discussing crystallographic structure and variations. Elemental analysis through ICP or spark. Authors have to point out the advantages of the proposed methodology, time, quantity of sample and costs compared to other methodologies by adding references. The citations are somewhat lacking in the discussion. In the Table 3 add a graph that facilitates the understanding of results.
Answer:
The reviewers gave us a good comment on the EDS analysis of Cr2O3 with the amounts of 2wt.%, 5wt.% and 10wt.% in the facecoat. But in this study, in order to compare the blades by using 2wt.% and 5wt.% h-BN added mould facecoats, only the blades by 2wt.% and 5wt.% Cr2O3 added mould facecoats were prepared and subsequently carried out the EDS analysis on the blade interface area. In addition, more Cr2O3 in the facecoat seems to be more likely to cause sand burning defects on the blades.
In the revised version, the EDS analysis results of blades prepared by adding Cr2O3 and h-BN to the mould facecoats have been given. At the same time, the characterization results of all blade interfaces have been discussed in more depth.
ICP and spark analysis are more accurate than EDS in element composition analysis of most of materials, but it is not suitable for element composition analysis of blade/moul interface layer in this study.
We have pointed out the advantages of the proposed mould facecoat by adding references that can be seen in the Introduction and Reference Parts in the revised vision. The citations have been further discussed in the revised manuscript.
Minor points:
1. Line 34 Reference 1 does not talk about nickel alloys review.
Answer: Reference 1 “Liang, X.F.; Zhao, Y.T Ma, Z.H.; Meng, X.F. Study on the Preparation and Microstructure of a Single-crystal Hollow Turbine blade. Mater. Manuf. Processes 2017, 32 (16), 1887-1892” in the original manuscript has been replaced by “Xiao, X.; Xu H.; Qin, X.Z.; Guo, Y.A.; Zhou, L.Z.; Guo, J.T. Study on thermal fatigue behavior of three kinds of nickel-based superalloy. Acta Metall. Sin. 2011, 28, 1129-1134” in the revised manuscript.
2. Line 41 Space in 30min ~ 90min, correct 30min ~ 90min.
Answer: Line 41 Space in 30min~90min in the original manuscript has been corrected as 30min ~ 90min in the revised manuscript. All the same errors have been corrected in the revised manuscript.
3. Line 46-59 Add references there are none.
Answer: We have rewritten the Introduction part. Some new references have been added in the revised manuscript.
4. Line 60-64 Add references there are none.
Answer: We have rewritten the Introduction part. Some new references have been added in the revised manuscript.
5. Table 4 homogenize decimals.
Answer: All the values are in the same decimal in Table 4 in the revised manuscript.
6. Line 102 space 24h. correct 24 h. reviewing the entire document is consistent this error.
Answer: All the same errors have been corrected in the revised manuscript.
The manuscript has been resubmitted to the journal. We look forward to your positive response.
Sincerely,
Dr. Fei Li
1 Shanghai Key Laboratory for High Temperature Materials and Precision Forming, Shanghai Jiao Tong University, Shanghai 200240,China
2 State Key Laboratory of Metal Matrix Composites, Shanghai Jiao Tong University, Shanghai 200240, China
Email address: [email protected]
Tel/Fax: +86-21-34202951

Round 2
Reviewer 3 Report
Based on the changes made, the manuscript can be accepted, considering these observations for future work.